# Prenatal Folate and Choline Levels and Brain and Cognitive Development in Children: A Critical Narrative Review

**DOI:** 10.3390/nu14020364

**Published:** 2022-01-15

**Authors:** Nathalie Irvine, Gillian England-Mason, Catherine J. Field, Deborah Dewey, Fariba Aghajafari

**Affiliations:** 1O’Brien Centre for the Bachelor of Health Sciences, Cumming School of Medicine, University of Calgary, 3330 Hospital Drive NW, Calgary, AB T2N 4N1, Canada; nathalie.irvine1@ucalgary.ca; 2Owerko Centre, Alberta Children’s Hospital Research Institute, University of Calgary, 2500 University Drive NW, Calgary, AB T2N 1N4, Canada; gillian.englandmason@ucalgary.ca (G.E.-M.); dmdewey@ucalgary.ca (D.D.); 3Department of Pediatrics, Cumming School of Medicine, Alberta Children’s Hospital, 28 Oki Drive NW, Calgary, AB T3B 6A8, Canada; 4Department of Agricultural, Food and Nutritional Science, University of Alberta, 4-126C Li Ka Shing Centre for Research, 11203-87th Ave NW, Edmonton, AB T6G 2H5, Canada; catherine.field@ualberta.ca; 5Hotchkiss Brain Institute, Health Research Innovation Centre, Cumming School of Medicine, University of Calgary, 3330 Hospital Drive NW, Calgary, AB T2N 4N1, Canada; 6Department of Community Health Sciences, Cumming School of Medicine, University of Calgary, 3D10, 3280 Hospital Drive NW, Calgary, AB T2N 4Z6, Canada; 7Department of Family Medicine, Cumming School of Medicine, University of Calgary, G012, 3330 Hospital Drive NW, Calgary, AB T2N 4N1, Canada

**Keywords:** pregnancy, choline, folate, children, neurodevelopment, brain development, cognitive development

## Abstract

Women’s nutritional status during pregnancy can have long-term effects on children’s brains and cognitive development. Folate and choline are methyl-donor nutrients and are important for closure of the neural tube during fetal development. They have also been associated with brain and cognitive development in children. Animal studies have observed that prenatal folate and choline supplementation is associated with better cognitive outcomes in offspring and that these nutrients may have interactive effects on brain development. Although some human studies have reported associations between maternal folate and choline levels and child cognitive outcomes, results are not consistent, and no human studies have investigated the potential interactive effects of folate and choline. This lack of consistency could be due to differences in the methods used to assess folate and choline levels, the gestational trimester at which they were measured, and lack of consideration of potential confounding variables. This narrative review discusses and critically reviews current research examining the associations between maternal levels of folate and choline during pregnancy and brain and cognitive development in children. Directions for future research that will increase our understanding of the effects of these nutrients on children’s neurodevelopment are discussed.

## 1. Introduction

A mother’s nutritional status, as defined by the intake and utilization of nutrients during pregnancy has a significant effect on the development of the fetal brain and subsequent cognitive outcomes later in life [1]. To support optimal fetal development and offspring cognitive function, it is recommended that pregnant women consume adequate, but not deficient or excessive, sources of nutrients (e.g., vitamin D, folate, choline, and iron), in addition to other food components (e.g., fibre, fat, and proteins) [2]. Notably, nutrient demands increase during pregnancy as maternal body mass and metabolic demands increase in conjunction with the development of the placenta and fetus [3,4]. Approximately 28 days after conception, the fetal neural plate folds and fuses, and forms the neural tube, which allows for the development of the fetal brain [5,6]. Certain neurodevelopmental processes, such as neural cell proliferation and cell migration, occur solely during gestation; however, other neurodevelopmental processes, such as neurogenesis, synapse formation, and myelination, begin during gestation and continue into adolescence and early adulthood [6]. Significant refinement of neural connections begins 24 weeks after conception and continues into the perinatal period [6,7]. Further, peak synapse development and significant brain growth in humans occurs between 34 weeks post conception and 2 years of age [7,8]. Maternal intake and utilization of nutrients provide the essential building blocks to support numerous cellular processes during fetal development, including cellular proliferation, DNA synthesis, and neurotransmitter and hormone metabolism [9,10,11,12]. Some nutrients, such as iron and long-chain polyunsaturated fatty acids, also enable axon myelination, synaptogenesis, and neurotransmitter transmission [9,12]. Therefore, the availability of nutrients to the developing fetus is likely to have long-lasting impacts on both the physical development of the brain as well as children’s cognitive development.

Cognition refers to the mental processes involved in the acquisition, retention, and use of knowledge, and the foundational aspects of cognitive development include attention, processing speed, representational competence (i.e., the ability to manipulate a mental image of an object or idea), and memory [13,14]. Nutritional research studies often assess children’s cognitive abilities using tests of attention, speed of information processing, learning and memory, executive functions (e.g., inhibitory control, cognitive flexibility, and working memory), and intelligence [15,16]. Children’s early cognitive performance on these types of assessments are predictive of later academic achievement and level of educational attainment [17,18,19]. Supporting healthy cognitive development in children begins in utero, as multi-micronutrient supplementation during the gestational period has been shown to be associated with improved cognitive outcomes in children at one and two years of age [20,21].

Two nutrients in particular, folate and choline, have been linked to the prevention of neural tube defects and to both health and cognitive outcomes in children [9,22,23,24,25,26]. Both folate and choline are methyl-donor nutrients, which means that they have been shown to alter DNA methylation and can have long-lasting impacts on gene expression and neuronal function; however, their effects on the development of children’s brains and cognition, and the mechanisms by which these effects occur have not been clearly established [9,27]. The objectives of this review are to discuss the current state of knowledge regarding the associations between maternal levels of folate and choline during pregnancy and children’s neurodevelopment, outline the limitations and knowledge gaps in the current literature, and suggest directions for future research. To address these objectives, we initially searched PubMed and Google Scholar for relevant articles between September 2020 and November 2020. In September 2021, we reran the search to determine if any additional relevant articles had been published. The keywords used were choline, folate, pregnancy, prenatal, child outcomes, cognitive outcomes, cognitive development, levels, concentrations, intake, mechanism, brain development, exposure, neurodevelopment, serum, plasma, red blood cell, food frequency questionnaires, supplementation, maternal, fetal development, offspring, infant, child, memory, intelligence, diet, nutrition, nutrients, dietary sources, deficiency, exposure, and gestational. The identified articles and relevant studies cited in the references of these articles were critically reviewed and information from 121 studies was included in this narrative review.

## 2. Gestational Folate and Fetal Brain Development

Folate, or vitamin B-9, is a water-soluble vitamin complex that is an essential nutrient [28]. It cannot be synthesized de novo by the body and must be obtained either from diet or supplementation [28]. Dietary folate is a naturally occurring nutrient found in foods such as leafy green vegetables, legumes, egg yolk, liver, and citrus fruit, whereas folic acid is a synthetic dietary supplement present in fortified foods and vitamins [29,30]. To increase the intake of folic acid in the general population and reduce the rate of neural tube defects, fortification of grain-based food products such as flour, cereal, and pasta with folic acid has been mandated in Canada and the United States since 1998 [30,31,32,33]. Currently, over 80 countries have mandated fortification of foods with folic acid, and it is recommended that men and non-pregnant women consume 320 to 400 μg of folate per day [30,34].

Compared to folate requirements of non-pregnant women, it is recommended that pregnant women consume at least 400 to 800 μg of folate per day to ensure healthy maternal, placental, and fetal tissue growth [34,35,36,37]. The placenta extracts folic acid from maternal circulation and concentrates it into fetal circulation [35,38,39,40]. This results in fetal levels that are two to four times higher than maternal levels [39,41,42,43,44,45]. In the fetus, folate is critically important for cellular proliferation, neural stem cell proliferation and differentiation, decreasing apoptosis, and altering and maintaining DNA synthesis [3,9,35]. Folate deficiency during pregnancy can result in megaloblastosis (i.e., large cells with arrested nuclear maturation) and cell death, particularly in rapidly proliferating somatic cells [35]. Embryonic neural tube and neural crest cells are highly proliferative and folate deficiency during the period of neural tube closure (21 to 28 days post conception) predisposes the fetus to neural tube defects [23,46].

Given the ethical considerations regarding maternal nutrition during pregnancy, experimental research examining the effects of maternal folate deficiency on fetal brain development has been conducted in animals only. Craciunescu et al. found that maternal folate intake affected fetal forebrain progenitor cells during late gestation [47,48]. In these studies, pregnant mice were all fed the same standard diet containing 2 mg/kg of folic acid until day 11 of gestation and then folic acid was removed from the diets of the mice in the folate-deficient group [47,48]. Offspring of folate-deficient mice were found to have fewer replicating neural progenitor cells in the hippocampus, caudate, putamen, striatum, anterior and mid-posterior neocortex, and ventricular zones of the septum [47,48]. It was also found that apoptotic cells were more prevalent in the fetal septum and hippocampus of mice from mothers fed a folate-deficient diet as compared to the control mice [47,48]. Folic acid supplements before and during early pregnancy are essential for preventing neural tube defects; however, these animal studies suggest that folate availability throughout gestation is critically important for fetal brain development in both cortical and subcortical regions of the brain. Figure 1 provides a schema representing the potential mechanism of action through which folate is associated with children’s brain and cognitive development.

## 3. Gestational Choline and Fetal Brain Development

Choline is also an essential nutrient but was only officially recognized as such in 1998 [49]. There are currently no mandates regarding choline supplementation of food in Canada or the United States [49,50,51]. Although choline is produced endogenously in the human liver, the amount of choline naturally produced is insufficient to meet the needs of the human body and thus adequate levels of choline can only be obtained through diet [52]. Animal food products such as beef, eggs, chicken, fish, and pork are major dietary sources of choline containing more than 60 mg of choline per 100 g [49,52,53,54]. Plant foods such as nuts, legumes, and cruciferous vegetables are also good dietary sources of choline containing at least 25 mg of choline per 100 g [53,54]. Choline can also be obtained through dietary supplements containing choline only, choline in combination with B-complex vitamins, or in some multivitamin products [54]. Choline dietary supplements typically contain between 10 mg and 250 mg of choline per dose and often provide choline in the form of choline bitartrate, phosphatidylcholine, or lecithin [54].

It is recommended that choline intake should increase from 425 mg/day in non-pregnant women to 450 mg/day in pregnant women [54]. Large amounts of choline from maternal diet are transferred across the placenta from the mother to the fetus [53,55,56,57]. Choline concentrations in the amniotic fluid are 10 times greater than in maternal blood, and plasma choline concentrations are 3 and 6 to 7 times greater in the umbilical cord and fetus, respectively, than they are in the maternal blood [44,45]. It is believed that choline plays a similar role to folate in brain development, including acting as a methyl donor in DNA methylation [9,51]. Like folate, choline is important for the prevention of neural tube defects as it is required for closure of the neural tube [9,22,58,59]. Choline also plays a significant role in hippocampal development and has been associated with the development of neural pathways and the expression of genes involved in memory processes [50,51].

In human diets, the most common dietary form of choline is phosphatidylcholine (PC); however, it can also be found as free choline, phosphocholine, sphingomyelin, glycerophosphocholine, and lysophosphatidylcholine [53]. Choline is a component of sphingomyelin, which is a constituent of the myelin sheath of nerve axons and facilitates efficient transmission of nerve signals [51,53]. As a methyl donor in DNA methylation, choline is thought to regulate the expression of genes involved in regulating synaptic plasticity, learning, and memory [50,51]. As with folate, choline also plays a role in the conversion of homocysteine to methionine and helps regulate homocysteine concentrations in the body [55]. Further, it acts as a major constituent of phospholipids in cell membranes and signaling lipids in cells and is involved in the biosynthesis of lipoproteins [9,51,53,55]. It is also a constituent of the neurotransmitter acetylcholine (ACh), which acts in both the peripheral and central nervous systems [9,53,55,60]. ACh influences processes in the developing brain including progenitor cell proliferation and differentiation, neurogenesis, gliogenesis, cell survival, morphology and migration, and synaptic plasticity, and supports the development of the hippocampus [53]. ACh released from cholinergic neurons that extend into the hippocampus and cerebral cortex from the basal forebrain also regulates multiple processes including attention, learning and memory [61].

Like experimental studies examining the effects of folate deficiency on brain development in mice, animal models have also shown that gestational choline levels have significant effects on fetal brain development. Studies have found that maternal choline deficiency resulted in decreased progenitor cell proliferation and neurogenesis, and increased apoptosis in the fetal hippocampus [56,62,63]. Further, maternal choline supplementation stimulated hippocampal cell division and enhanced hippocampal neurogenesis [56,64]. Other studies have found that modification of choline intake levels in pregnant rats resulted in changes in the timing of migration and differentiation of neuronal progenitor cells in fetal hippocampal regions [62,65]. Thus, to support healthy brain differentiation and development, particularly in regions important for learning and memory such as the hippocampus, research suggests that it is important for a fetus to have access to adequate amounts of choline throughout gestation. Figure 2 provides a schema representing the potential mechanism of action through which choline is associated with brain and cognitive development in children.

## 4. Potential Interactions between Gestational Folate and Choline

Few studies have examined the interactive effects of folate and choline on fetal brain development. Craciunescu et al. found choline supplementation may modify the effects that dietary folate availability has on neural progenitor cells in the fetal forebrain during late gestation [48]. Specifically, they reported that the rate of mitosis in folate-deficient choline-supplemented (FDCS) mice (i.e., 4.95 g choline chloride/kg diet, 0.0 mg folic acid/kg diet) was less than that of control mice who were supplemented with standard amounts of folate and choline (i.e., 1.1 g choline chloride/kg diet, 2 mg folic acid/kg diet) but greater than that of folate-deficient mice (i.e., 1.1 g choline chloride/kg diet, 0.0 mg folic acid/kg diet) [48]. Further, hippocampal apoptosis rates in FDCS mice were found to be significantly lower than those of folate-deficient mice, but the same as those in control mice [48]. As both folate and choline are methyl-donor nutrients that influence neurogenesis and apoptosis, it is possible that choline supplementation might mitigate the effects of folate deficiency on brain development, and this may be mediated by epigenetic events (i.e., methylation of DNA and histones important for epigenetic control of gene expression) [66,67]. Investigations of the interactive effects of folate and choline is important because they could provide a better understanding of how these nutrients work together to support brain and cognitive development. Future research may also provide further evidence that maternal gestational supplementation of these two nutrients is critical for optimal fetal brain development and subsequent cognitive outcomes in children.

## 5. The Current State of Knowledge on the Association between Maternal Folate and Offspring Cognitive Outcomes

Animal studies (see Table 1) suggest a relationship between maternal folate intake during pregnancy and offspring brain development; however, a limited number of studies have examined the effects of maternal folate levels on cognition. Jadavji et al. found that maternal folate deficiency in mice (i.e., 0.3 mg folic acid/kg diet) was associated with short-term memory impairment in offspring based on their poorer performance on the novel object recognition task and the Y-maze test compared to offspring of mice fed folate sufficient diets (i.e., 2 mg folic acid/kg diet) [68]. Additionally, Ferguson et al. found that maternal folate deficiency (e.g., 400 nmol of folic acid/kg diet) in mice produced offspring who exhibited more anxiety-related behavior on the elevated plus maze test [69]. These animal studies suggest that maternal gestational folate deficiency is associated with poorer cognitive and behavioural outcomes in offspring and that gestational folate availability is important for offspring cognitive and behavioural development.

Human studies (see Table 1) have also examined the association between maternal folate levels during pregnancy and child cognitive outcomes; however, various methods have been used to measure maternal folate levels. Several studies considered maternal use of folic acid supplements only during pregnancy. Julvez et al. found that in four-year-old children, verbal, motor- executive function, and verbal-executive function scores on the McCarthy Scales of Children’s Abilities, and social competence and inattention symptom scores on the California Preschool Social Competence Scale were positively associated with maternal use of folic acid supplements during the first trimester of pregnancy [70]. Similarly, Wehby and Murray assessed the effect of using folic acid supplements at conception and/or during the first trimester of pregnancy on cognitive development at three years of age using 16 items from the Denver Developmental Screening Test and found that prenatal folic acid supplementation had a positive effect on children’s overall cognitive and gross motor development [71]. A study conducted in the United Kingdom reported that seven-year-old children of mothers who received 400 μg folic acid per day during the second and third trimester of pregnancy had higher scores on the Wechsler Preschool and Primary Scale of Intelligence, Third Edition (WPPSI-III) as compared to the children of mothers who were given a placebo [72]. Further, compared to a nationally representative sample of children seven years of age, the children of the folate-supplemented mothers had higher verbal, performance, general language, and Full-Scale IQ scores on the WPPSI-III [72]. It is notable that the folic acid supplement given to the women in this study was at the low end of the recommended range for folic acid intake for pregnant women (e.g., 400–800 μg/day) [34,36,37,72,73]. In another study, Chatzi et al. examined neurodevelopmental outcomes on the Bayley Scales of Infant and Toddler Development, Third Edition (Bayley-III) in 18-month-old children of mothers who reported taking daily supplements of 5000 μg of folic acid or more throughout pregnancy compared to children of mothers who did not use a folic acid supplement throughout pregnancy [74]. They found that children of mothers who reported taking a daily supplement had a 5 unit increase on receptive communication and a 3.5 unit increase in expressive communication on the Bayley-III scales [74]. The maternal folic acid intake of the women in this study was well above the recommended intake for pregnant women (e.g., 400–800 ug/day) [34,36,37,73,74]. Finally, a recent meta-analysis conducted by Chen et al., which investigated the effect of maternal folic acid supplementation on children’s neurodevelopment, concluded that appropriate maternal folic acid supplementation may have positive effects on children’s intelligence and development and reduce the risk of language problems, ADHD, autism traits, and behavioral problems [75]. Cumulatively, the research evidence suggests that maternal folic acid supplementation, even at the lower end of the recommended range throughout pregnancy, is an important predictor of children’s cognitive outcomes. However, in the majority of these studies, women’s actual folate levels were not measured and folate intake from their diets was not considered.

In addition to the above supplementation studies, other studies have used food frequency questionnaires (FFQs) to estimate levels of maternal prenatal folic acid and examine associations with cognitive outcomes. For example, Villamor et al. found that for each 600 μg per day increase in total folate intake during the first trimester of pregnancy assessed using FFQs there was a 1.6-point increase in scores on the Peabody Picture Vocabulary Test, Third Edition (PPVT-III) in three-year-old children [76]. The women in this study had a mean estimated folate intake of 949 ± 390 μg/day, which was well above the recommended daily intake for pregnant women [76]. In contrast, Boeke et al. did not find an association between increased folate intake assessed using FFQs during the first and second trimesters of pregnancy and higher scores on the Wide Range Assessment of Memory and Learning, Second Edition (WRAML2) or the Kaufman Brief Intelligence Test, Second Edition (KBIT-2) in children at age seven [77]. The women in this study had a mean daily estimated folate intake of 972 ± 392 μg/day and 1268 ± 381 μg/day in the first and second trimesters of pregnancy, respectively, which were well above the recommended intake levels for pregnant women [77]. The results of these studies suggest that maternal folic acid intake may be an important predictor of some areas of cognitive development (e.g., language) but not of other areas (e.g., memory and intelligence). The use of FFQs to quantify maternal folic acid intake accounts for most dietary sources of folic acid and thus provides an accurate estimation of women’s actual folate intakes; however, there can be variability among FFQs in the dietary sources of folic acid considered and participants’ recall can limit the accuracy of this method [78,79].

Studies have also measured maternal folate concentrations (i.e., plasma/serum folate, red blood cell (RBC) folate) directly from blood samples collected during pregnancy. Although plasma/serum and RBC folate concentrations are both associated with folic acid intake [20,21], they result from different biologic processes and are not interchangeable [80,81]. Plasma/serum folate concentrations reflect very recent intake, whereas RBC folate concentrations reflect the long-term average of intake over the life span of red blood cells and folate stores in the liver [82]. As noted above, it is recommended that pregnant women should consume at least 400 to 800 μg of folate per day; however, a corresponding plasma/serum folate level has not been established [34,35,36,37,83,84]. Therefore, the results of studies that have examined the associations between plasma/serum and RBC folate levels in maternal blood and child outcomes may not be comparable. A study in India that measured maternal plasma/serum folate concentrations at 28 to 32 weeks’ gestation (i.e., third trimester) reported a positive association with children’s performance on a test of cognitive function (i.e., Kaufman Assessment Battery for Children) [85]. The women in this study had mean plasma/serum folate concentrations of 34.7 ± 19.2 nmol/L, which were within the normal range (i.e., 7–46 nmol/L) for non-pregnant women [34,85,86,87]. Ars et al. found that low plasma/serum folate concentrations below 8 nmol/L during the first trimester of pregnancy were associated with smaller total brain volume and poorer language and visuospatial skills on the NEPSY-II-NL in children at six years of age [88]. In contrast, maternal plasma/serum folate concentrations at 16 and 36 weeks gestation (i.e., second and third trimesters) were not associated with scores on the Bayley-III at 18 months of age [89]. The women in this study had mean plasma/serum folate concentrations of 36.4 ± 8.08 nmol/L at 16 weeks gestation, which was in the normal range for non-pregnant women [34,86,87,89]. Tamura et al. examined the association between RBC folate status during the second and third trimesters of pregnancy and children’s neurodevelopment at age five years and found no associations [90]. Neurodevelopment was assessed using the Differential Ability Scales, Visual and Auditory Sequential Memory Tests, Knox Cube Test, Gross Motor Scale, and Grooved Pegboard Test [90]. In this study, the women’s mean RBC folate concentrations were 873 nmol/L, 1070 nmol/L, and 1096 nmol/L at 19, 26, and 37 weeks gestation, respectively, which were slightly above or slightly below the recommended RBC folate concentration of 906 nmol/L for pregnant women [34,37,90]. Like studies that used FFQs, which estimate recent folate levels, there was some support that maternal plasma/serum folate levels were associated with children’s cognitive outcomes. However, both FFQs and plasma/serum folate assess a woman’s folate levels at a given time point. They are not predictive of folate levels across pregnancy as levels may vary depending on when the blood sample was taken [91,92,93]. In contrast, the Tamura et al. study, which investigated the association between RBC folate (which is reflective of longer-term intake), and child outcomes did not report any associations, suggesting that the inconsistencies in the findings of human studies may be due to differences in the measure used to assess folate levels.

## 6. The Current State of Knowledge on the Associations between Maternal Choline and Offspring Cognitive Outcomes

In rat and mice models (see Table 2), high maternal choline intake during pregnancy has been found to enhance the cognitive abilities of offspring; however, the optimal amount is not known [50]. Pregnant rats whose diet was supplemented with choline chloride were found to produce offspring with enhanced visuospatial memory skills as compared to non-choline-supplemented rats [94]. Choline supplementation in pregnant rats has also been linked to increased ability to retain a larger number of items in working memory, increased memory precision due to less proactive interference, an increased tendency for object exploration, and increased thresholds for implementing chunking strategies in offspring [94,95,96,97]. Further, choline supplementation in pregnant rats facilitates temporal memory and accelerates the maturation of relational cue processing in offspring [51,98]. These findings suggest that choline supplementation during pregnancy may be predictive of better cognitive function in offspring, particularly offspring memory.

Enhanced cognitive function associated with increased maternal intake of choline during gestation has also been noted in human studies (see Table 2). The methods used to investigate choline levels in women were like those used in studies that examined maternal folate levels (i.e., supplementation, FFQs, measurements from blood samples). One study reported that infants born to women who were supplemented with 930 mg of choline chloride per day, compared to infants of women who were receiving 480 mg of choline per day, displayed improved information processing speed measured using a visual attention task [99]. It is of note that the choline levels in both groups were above adequate intake levels for pregnant women [34,54,99]. Another study used a FFQ to estimate choline intake during the first and second trimesters and found that prenatal choline intake was positively associated with memory scores on the WRAML2, but not intelligence scores on the KBIT-2 in seven-year-old children [77]. The women in this study had a mean estimated daily choline intake of 328 ± 63 mg/day, which was below the recommended adequate intake level for pregnant women (i.e., 450 mg/day) [34,54,77]. A study that directly measured maternal plasma-free choline at 16 weeks gestation reported positive associations with infant cognitive development at 18 months of age on the Bayley-III [89]. The mean plasma-free choline concentration of the women was 7.07 ± 1.78 umol/L and the mean estimated daily choline intake was 383 ± 98.6 mg/day at 16 weeks gestation, which was below the recommended adequate intake level for pregnant women [34,54,89]. However, in a study conducted by Signore et al., maternal gestational free and total serum choline concentrations measured at multiple timepoints throughout pregnancy (i.e., at 16 to 18 weeks, 24 to 26 weeks, 30 to 32 weeks, and 36 to 38 weeks gestation) were not associated with Wechsler Preschool and Primary Scales of Intelligence-Revised (WPPSI-R) Full-Scale IQ, visuospatial processing or memory in children at five years of age [100]. There are no recommendations regarding serum choline concentrations in maternal blood during pregnancy; therefore, no conclusions can be drawn as to whether these women were below, within, or above adequate recommended serum choline levels for pregnant women. The results from these studies suggest that maternal choline may be predictive of children’s memory, attention, and processing speed, even at low levels; however, future research is needed that examines these relationships when choline intake during pregnancy is at or above recommended intake levels. Additionally, the form of choline measured in studies should also be considered as one study found that plasma free choline concentrations were not predictive of dietary choline intake during pregnancy [101]. Considering the evidence from animal models supporting the importance of maternal choline for brain development and particularly hippocampal function, and that choline supplementation might mitigate the negative effects of folate deficiency on brain development, future research that examines folate and choline together and considers possible interactions between these two nutrients is needed.

## 7. Gaps in the Current State of Knowledge and Directions for Future Research

Animal studies have established that gestational availability of folate and choline influences numerous processes that are critical for healthy development of the fetal brain. The current literature also suggests that maternal folate and choline levels during pregnancy may be important predictors of children’s cognitive outcomes; however, there are inconsistencies in the findings that have yet to be resolved. These inconsistencies may be a result of several methodological constraints and confounding factors that have not been accounted for in the current literature.

One such methodological constraint is that the methods used to assess folate and choline levels in the current literature are not consistent across studies and some have inherent limitations that limit validity of the findings. For example, several studies have simply assessed whether women did or did not take folic acid or choline supplements during pregnancy and did not account for dietary sources of folate or choline [70,71,72,74,99]. FFQs were also used frequently to estimate women’s folate and choline intake; however, studies have shown that there can be significant variability between the dietary sources of these nutrients that are considered, and levels of these nutrients in the dietary sources [78,79]. This variability could influence the consistency of the results across studies that use this method. Other studies have assessed plasma/serum levels of folate or choline concentrations; however, there are currently no recommendations regarding their adequate or recommended levels in pregnant women [85,89,100]. Thus, classifying pregnant women as below, within, or above recommended ranges for these nutrients during pregnancy based on their concentrations in serum/plasma blood samples is not currently possible. The use of various methods to assess women’s folate and choline levels during pregnancy limit the possibility of comparing the results of the research conducted to date. Further, the lack of consistency in the findings reported in the human literature could be due to the different methods used to measure folate and choline levels. Future research is needed that investigates the biological relevance of folate and choline intake and plasma/serum levels and RBC folate levels on human brain development and their association with children’s cognitive outcomes.

Another potential methodological constraint is that the gestational trimester in which folate and choline levels are measured could be differentially associated with children’s cognitive outcomes [70]. Changes in concentrations of folate and choline throughout pregnancy and the impact of varying concentrations of these nutrients on children’s neurodevelopmental outcomes are areas of research in need of further investigation. Lewis et al. and Signore et al. both found that maternal free and total choline concentrations remain relatively constant throughout gestation [100,102]. However, Fayyaz et al. found that folate concentrations tend to increase throughout pregnancy [86]. Interestingly, Villamor et al. found that maternal folate intake during the first trimester, but not the second trimester of pregnancy, was associated with children’s scores on the PPVT-III at three years of age, suggesting that the timing of folate supplementation during pregnancy could influence children’s neurodevelopment [76]. However, the precise timing and the ideal concentrations of these nutrients at different time points during pregnancy to ensure optimal neurodevelopment are currently unknown.

Another issue of concern is that the research to date has examined a limited number of potential confounders. Demographic factors such as family socioeconomic status and maternal and paternal level of educational attainment have been associated with maternal nutrient use and children’s cognitive development [103,104]. Sample characteristics such as maternal pre-pregnancy body mass index (BMI) and child sex have also been found to be confounding factors that can influence children’s cognitive development [105,106]. Finally, other important micronutrients that are often supplemented during pregnancy such as vitamins B-12 and B-6, omega-3 long-chain polyunsaturated fatty acids, and iron have also been found to be associated with children’s cognitive outcomes [107,108,109,110,111,112,113,114,115]. These variables need to be considered as potential confounding factors when assessing the effects of folate and choline on children’s neurodevelopment.

In addition to the already identified methodological and analytical issues, it is important to note that there is an imbalance in the number of studies that considered the effects of maternal folate compared to maternal choline. More human studies have examined the association between maternal folate levels and children’s cognitive outcomes and no studies have investigated their interactive effects. The evidence from animal models suggesting that gestational choline supplementation may reduce the adverse effects of folate deficiency on fetal brain development, supports the conduct of human studies that investigate how these two nutrients interact to influence cognitive outcomes in children.

Further, limited research has investigated possible biological mechanisms (e.g., DNA methylation alterations) that may underlie these associations. It has been noted that folate and choline likely play similar roles in the development of the fetal brain and central nervous system and that their interaction may influence fetal brain development [9,48]. However, to date, no human studies have examined potential synergistic effects of maternal folate and choline levels during pregnancy and potential underlying biological mechanisms (i.e., DNA methylation) on children’s cognitive outcomes

One last point that needs to be considered in future research is the level of supplementation. Several studies have reported that supplementing with large amounts of folate or choline during pregnancy may result in adverse child health outcomes, including low birth weight and increased risk for developing autism spectrum disorder (ASD) or colitis [116,117,118,119,120,121]. However, there is little research on the association between very high maternal choline or folate supplementation and adverse child neurodevelopmental outcomes. Further, the precise levels above the daily recommended levels for folate and/or choline that would be harmful to children’s neurodevelopment have yet to be determined.

In summary, the findings from this narrative review suggest that future nutritional research examining the associations between maternal folate and choline levels and children’s cognitive outcomes should develop and use consistent and accurate methods to measure women’s folate and choline levels across the different trimesters of pregnancy and consider a wider range of potential confounding variables (e.g., demographic factors, levels of other nutrients) in order to reduce inconsistencies in results that could be attributable to discrepancies in methodology. Additionally, future research is needed that addresses the following issues: (a) What are the optimal intake levels/status of folate and choline needed during pregnancy to promote optimal child neurodevelopment? (b) Do the optimal levels/status vary across the different trimesters of pregnancy? and (c) Is there a risk to children’s neurodevelopment if maternal intake/status is above or below the optimal levels? Additionally, potential biological pathways (e.g., DNA methylation) should be examined to further understand the mechanistic basis for both folate’s and choline’s role in children’s brain and cognitive development and how these nutrients might interact within the context of early development to further support children’s neurodevelopment. It is important that future research address these methodological limitations and mechanistic questions to better understand how maternal folate, choline, and their interaction during pregnancy, influence brain and cognitive development in children.

## 8. Conclusions

Folate and choline are important methyl-donor nutrients that play critical roles in the development of the fetal brain. Animal studies indicate that gestational folate and/or choline levels are associated with structural and cellular alterations in various brain regions such as the cerebral cortex and hippocampus. The current human literature also suggests that maternal folate and choline concentrations during pregnancy may play a role in children’s cognitive development; however, there are some inconsistencies in published research. To address these inconsistencies, future research needs to use consistent methods to assess women’s folate and choline levels during pregnancy, and consider confounding factors (e.g., parental socioeconomic status, maternal pre-pregnancy BMI, infant birth weight, maternal levels of other micronutrients during pregnancy, gestational trimester), relative maternal folate and choline concentrations (e.g., below, at, or above recommended levels) and the potential synergistic effects that maternal folate and choline levels may have on fetal brain development and in turn children’s cognitive development.

## Figures and Tables

**Figure 1 nutrients-14-00364-f001:**
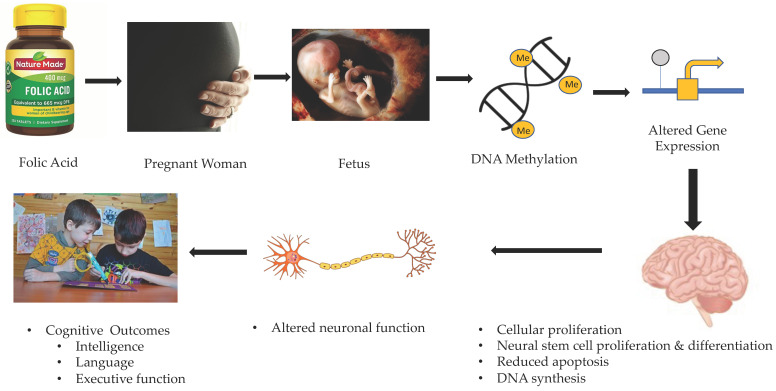
Potential mechanism of action through which prenatal folate affects brain and cognitive outcomes in children.

**Figure 2 nutrients-14-00364-f002:**
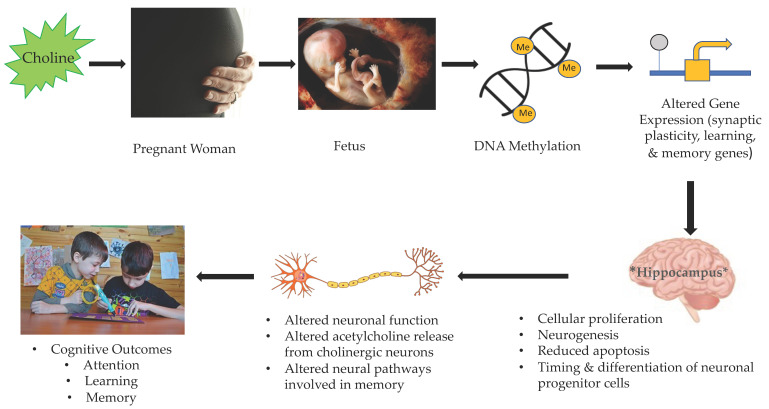
Potential mechanism of action through which prenatal choline affects brain and cognitive outcomes in children.

**Table 1 nutrients-14-00364-t001:** Characteristics of the animal and human studies that examined associations between maternal prenatal folate and offspring cognitive outcomes.

Identification	Location	Sample	Method Used to Determine Prenatal Folate Level	Maternal Folate Assessment	Dose of Folic Acid Intake or Mean Levels in Blood	Offspring’s/Children’s Age at the Time of Assessment	Offspring’s/Children’s Assessment	Main Results
**Animal Studies**								
[48] Craciunescu et al., 2010	United States	Fetal mice	Diet supplementation	Embryonic days 11–17	Control diet: 2 mg folic acid/kg diet; 1.1 g choline chloride/kg diet	Embryonic day 17	Histological and immune-histochemical assays of fetal brain regions of interest	Folate-deficient mice had fewer neural progenitor cells undergoing mitosis in the septum and greater apoptosis rates in the septum and hippocampus compared to the control mice
					Folate-deficient diet: 10.0 mg folic acid/kg diet, 1 g choline chloride/kg diet			
					Folate-deficient choline-supplemented diet: 0.0 mg folic acid/kg diet, 4.95 g choline chloride/kg diet			
[68] Jadavji et al., 2015	Canada	Methylenetetrahydrofolate reductase (MTHFR)-deficient male mice	Diet supplementation	6 weeks prior to breeding until the end of lactation	Folate-deficient diet: 0.3 mg folic acid/kg diet	3-week-old male mice	Novel object recognition task, Y-maze task	Maternal folate deficiency in mice was associated with short-term memory impairment in offspring
					Folate sufficient diet: 2 mg folic acid/kg diet			
[69] Ferguson et al., 2005	United States	20 female mice per group. 4 males and 4 females/litter were retained for initial behavioral testing. 1 male and 1 female pup/litter were used in postweaning behavioral testing	Diet supplementation	8 weeks prior to breeding until the end of gestation	Folate-deficient diets: (1) 400 nmol folic acid/kg diet and (2) 600 nmol folic acid/kg diet	Postnatal days 4–83	Righting reflex, negative geotaxis, forelimb hanging, motor coordination using Rotarod apparatus, open field activity, elevated plus maze	Maternal folate deficiency in mice produced offspring who exhibited more anxiety-related behavior in the elevated plus maze
					Folate sufficient diet: 1200 nmol folic acid/kg diet			
**Human Studies**								
[70] Julvez et al., 2009	Spain	420 mother-child pairs	Maternal self-report questionnaire used to determine whether supplements containing folic acid were taken	First trimester	No or yes	4 years of age	McCarthy Scales of Children’s Abilities, California Preschool Social Competence Scale	Verbal, motor-executive function, verbal-executive function, social competence and inattention symptom scores were positively associated with maternal use of folic acid supplements
[71] Wehby and Murray 2008	United States	6774 mother-child pairs	National survey that included data as to whether supplements containing folic acid had been taken	First trimester	No or yes	3 years of age	16 items from the Denver Developmental Screening Test	Prenatal folic acid supplementation had a positive effect on children’s overall cognitive and gross motor development
[72] McNulty et al., 2019	United Kingdom	37 mother-child pairs in the treatment group and 33 mother-child pairs in the placebo group	Supplements or placebos distributed to the women in 7-day pillboxes	14 weeks gestation until the birth of the child	400 ug/day supplement containing folic acid	7 years of age	Wechsler Preschool and Primary Scale of Intelligence, Third Edition, UK Edition (WPPSI-III)	Children of mothers who were in the treatment group had higher scores on the WPPSI-III compared to the children of mothers who were given a placebo
					Placebo containing no folic acid			
[74] Chatzi et al., 2012	Greece	553 mother-child pairs	A questionnaire was administered by a trained research nurse that asked whether the women had taken a folic acid supplement since they became pregnant. Supplement users reported the brand name, the dose, and the frequency of intake, which was converted into a measure of daily intake.	14–18 weeks gestation	No folic acid intake from supplements	18 months of age	Bayley Scales of Infant and Toddler Development, Third Edition (Bayley-III)	Children of mothers who reported taking a daily supplement of 5 mg of folic acid or more had a 5 unit increase on receptive communication and a 3.5 unit increase in expressive communication
					Daily intake of 5 mg of folic acid from supplements			
					Daily intake of folic acid from supplements higher than 5 mg			
[75] Chen et al., 2021	China	32 cohort studies and 7 case–control studies	Systematic review and meta-analysis of research articles that examined the association between prenatal folic acid supplementation and postnatal neurodevelopmental outcomes. All studies discussed folic acid supplementation only	During pregnancy	Varied by study; some reported whether or not women were supplemented; others reported specific supplementation levels.	18 months to 17 years	Varied by study; different measures used to assess intelligence, risk of autistic traits, ADHD, behavior, language, and psychomotor problems	Appropriate maternal folic acid supplementation may have positive effects on children’s intelligence and development, and reduce the risk of autism traits, ADHD, and behavioral and language problems
[76] Villamor et al., 2012	United States	1210 mother-child pairs	Food frequency questionnaires	First and second trimester	Mean estimated folate intake of 949 ± 390 ug/day	3 years of age	Peabody Picture Vocabulary Test, Third Edition (PPVT-III), Wide Range Assessment of Visual Motor Abilities	For each 600 μg per day increase in total folate intake during the first trimester of pregnancy there was a 1.6-point increase in scores on the PPVT-III in the children
[77] Boeke et al., 2013	United States	895 mother-child pairs	Food frequency questionnaires	First and second trimesters	Mean daily estimated folate intake of 972 ug/day in the first trimester and 1268 ug/day in the second trimesters	7 years of age	Wide Range Assessment of Memory and Learning, Second Edition (WRAML2), Kaufman Brief Intelligence Test, Second Edition (KBIT-2)	No associations were found between maternal folate intake and child cognitive outcomes
[85] Veena et al., 2010	India	536 mother-child pairs	Maternal plasma/serum folate concentrations from blood samples	28–32 weeks gestation	Mean plasma/serum folate concentrations of 34.7 nmol/L	9 or 10 years of age	Kaufman Assessment Battery for Children	Positive association between maternal plasma/serum folate concentrations and children’s performance on the Kaufman Assessment Battery for Children
[88] Ars et al., 2019	Netherlands	256 mother-child pairs (62 in the low folate group and 194 in the normal folate group)	Maternal plasma/serum folate concentrations from venous blood samples	First trimester	Low folate group: plasma/serum folate levels below 8 nmol/L	6 years of age	Neuroimaging and NEPSY-II-NL	Low maternal plasma/serum folate concentrations below 8 nmol/L were associated with smaller total brain volume and poorer language and visuospatial skills in children
					Normal folate group: plasma/serum folate levels above 8 nmol/L			
[89] Wu et al., 2012	United States	154 mother-child pairs	Food frequency questionnaires and maternal plasma/serum folate concentrations from blood samples	16 weeks gestation	Mean plasma/serum folate concentrations of 36.4 nmol/L	18 months of age	Bayley Scales of Infant Development, Third Edition (Bayley-III)	No association found between maternal plasma/serum folate concentrations and child cognitive function
[90] Tamura et al., 2005	United States	355 mother-child pairs	Maternal red blood cell folate concentrations from blood samples	Second and third trimesters	Mean red blood cell folate concentrations were 873 nmol/L, 1070 nmol/L, and 1096 nmol/L at 19, 26, and 37 weeks gestation, respectively	5 years of age	Differential Ability Scales, Visual and Auditory Sequential Memory Tests, Knox Cube Test, Gross Motor Scale, and Grooved Pegboard Test	No association found between maternal red blood cell folate concentrations and child cognitive function

**Table 2 nutrients-14-00364-t002:** Characteristics of the animal and human studies that examined the associations between maternal prenatal choline and offspring cognitive outcomes.

Identification	Location	Sample	Method Used to Determine Prenatal Choline Level	Maternal Choline Assessment	Dose of Choline Intake or Mean Levels in Blood	Offspring’s/Children’s Age at the Time of Assessment	Offspring’s/Children’s Assessment	Main Results
**Animal Studies**								
[94] Meck et al., 1988	United States	16 male albino rats from 8 pregnant female rats	Diet supplementation	2 days prior to conception until birth of offspring	Control diet: 50 mM saccharin	60 days of age	12 and 18 arm radial maze task	Offspring of choline-supplemented rats had enhanced visuospatial memory skills as compared to offspring of non-choline-supplemented rats
					Choline supplemented diet: 50 mM saccharin containing 5 mL/L choline chloride			
[95] Glenn et al., 2008	United States	20 rats (10 supplemented, 10 not supplemented)	Diet supplementation	Embryonic days 12–17	Control diet: 1.1 g/kg choline chloride	1 and 24 months of age	Open field exploration, novel object exploration, BrdU immuno-histochemistry and unbiased stereology to assess hippocampal plasticity markers	Prenatal choline supplementation was positively associated with exploratory behavior in rats and preserved some features of hippocampal plasticity in offspring over a 2-year time span
		8 male and 8 female pups from each experimental group were used in the behavioral tests			Choline-supplemented diet: 5 g/kg choline chloride			
[96] Meck and Williams 1999	United States	30 offspring of 18 pregnant rats	Supplementation in drinking water	Embryonic days 11–18	Choline-deficient diet: no choline or saccharin	120 days of age	Twelve arm radial maze	Choline-supplemented and choline-deficient rats performed more accurately than control rats during spaced trials. Choline-supplemented rats displayed less proactive interference during massed trials compared to control and choline-deficient rats
					Control diet: 50 mM saccharin			
					Choline-supplemented diet: 50 mM saccharin and 25 mM choline chloride			
[97]. Meck and Williams 1997	United States	34 pregnant dams that produced 128 adult female rats	Supplementation in drinking water	Embryonic days 12–17	Control diet: 50 mM saccharin	60 days of age	6, 12, 18, 24 radial arm mazes	Rats treated perinatally with choline had a higher threshold for implementing a chunking strategy in the radial arm maze tasks
					Supplemented diet: 50 mM saccharin and 25 mM choline chloride			
[98] Meck and Williams 1997	United States	30 offspring of 18 pregnant rats	Supplementation in drinking water	Embryonic days 11–18	Choline-deficient diet: no choline or saccharin	4–6 months of age and 24–26 months of age	Peak-interval timing procedure	Prenatal choline supplementation was positively associated with cognitive function in offspring and choline deficiency was positively associated with impaired divided attention and accelerated age-related declines in temporal processing
					Control diet: 50 mM saccharin			
					Choline-supplemented diet: 50 mM saccharin and 25 mM choline chloride			
**Human Studies**								
[77] Boeke et al., 2013	United States	895 mother-child pairs	Food frequency questionnaires	First and second trimesters	Mean estimated daily choline intake was 328 mg/day	7 years of age	Wide Range Assessment of Memory and Learning, Second Edition (WRAML2), Kaufman Brief Intelligence Test, Second Edition (KBIT-2)	Prenatal choline intake was positively associated with memory scores, but not intelligence scores
[89] Wu et al., 2012	United States	154 mother-child pairs	Food frequency questionnaires; measured maternal plasma-free choline concentrations from blood samples	16 weeks gestation	Mean plasma-free choline concentration of the women was 7.07 umol/L; mean estimated daily choline intake was 383 mg/day	18 months of age	Bayley Scales of Infant Development, Third Edition (Bayley-III)	Positive associations were found between maternal plasma free choline concentrations during pregnancy and infant cognitive development
[99] Caudill et al., 2018	United States	12 mother-child pairs in the control group and 12 mother-child pairs in the treatment group	Supplement mixed in juice consumed at the study facility	27 weeks gestation until the birth of the offspring	480 mg choline/day or 930 mg choline/day	∼4, 7, 10, and 13 months of age	Visual attention task	Infants born to women supplemented with 930 mg of choline chloride per day compared to infants of women, who received 480 mg of choline per day, displayed higher information processing speed
[100] Signore et al., 2008	United States	404 mother-child pairs	Maternal free and total serum choline concentrations from blood samples	16 to 18 weeks, 24 to 26 weeks, 30 to 32 weeks, and 36 to 38 weeks gestation	Median free choline concentrations increased from 9.34 to 10.10 umol/L over the gestational period; median total choline concentrations increased from 2.57 to 2.75 mmol/L over the gestational period	5 years of age	Wechsler Preschool and Primary Scales of Intelligence-Revised (WPPSI-R)	No associations found between maternal prenatal choline concentrations and offspring WISC-R Full-Scale IQ, visuospatial processing, or memory

## Data Availability

Not applicable.

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
