# Peer review of "Prenatal Folate and Choline Levels and Brain and Cognitive Development in Children: A Critical Narrative Review"

_nutrients, 2022, doi:10.3390/nu14020364_

Round 1

Reviewer 1 Report

  Very up-to-date, extremely comprehensive manuscript describing the importance of two supplements of a pregnant diet well known. folate and less known choline Both are very important during pregnancy and could influence brain and cognitive development.

These nutrients have been linked to the prevention of neural tube defects and to both health and cognitive outcomes in children. A study conducted in the United Kingdom reported 226 that seven-year-old children of mothers who received 400 ug folic acid per day during the 227 second and third trimester of pregnancy had higher scores on the Wechsler Preschool and 228 Primary Scale of Intelligence The authors searched the database of PubMed and Google Scholar for relevant articles.

However, they did not indicate how many articles they based their knowledge on the authors described the mechanisms of action of folate and choline which are methyl-donor nutrients, which means that they have been shown to alter DNA methylation.

Authors describe a potential Interactions Between Gestational Folate and Choline An interesting aspect is provided by experimental studies showing hippocampal apoptosis in foliate mice was significantly lower than those of folate-deficient mice. it is possible that choline supplementation might mitigate the effects of folate deficiency on brain development, and this may be mediated by epigenetic events.

Other research novelty indicates that choline supplementation might mitigate the effects of folate deficiency on brain development, and this may be mediated by epigenetic events   . The article provides the recommended doses of both supplements for non-pregnant and pregnant women. The article disseminates the information that currently, over 80 countries have mandated fortification of foods with folic acid and. The described preferential transport of choline through the placenta is also of great importance for perinatologists as well as neonatologists.  This preferential active transport is pictured 10 times greater than in maternal blood plasma concentration. 

Author Response

Very up-to-date, extremely comprehensive manuscript describing the importance of two supplements of a pregnant diet well known. folate and less known choline Both are very important during pregnancy and could influence brain and cognitive development.

We thank the reviewer for their positive comments regarding this work.

The authors searched the database of PubMed and Google Scholar for relevant articles. However, they did not indicate how many articles they based their knowledge on.

We do not have an exact count of the number of articles that we identified during our search but the number of articles included in this review (N = 121), were the bases of our knowledge as per this review. This is indicated in line 115-117.

Reviewer 2 Report

The objectives of this review are to discuss the current state of knowledge regarding the associations between maternal levels of folate and choline during pregnancy and children’s neurodevelopment, outline the limitations and knowledge gaps of the current literature, and suggest directions for future research.

Comments and Suggestions for Authors:

The manuscript is an interesting narrative review, but requires some considerations.

Title:  You should make it explicit that this is a narrative review.

  1. Introduction.

Page 2, line 84. The search criteria used should be exposed: dates, keywords...

  1. Gestational Folate and Fetal Brain Development.

This meta-analysis should have been discussed:

Chen H, Qin L, Gao R, Jin X, Cheng K, Zhang S, Hu X, Xu W, Wang H. Neurodevelopmental effects of maternal folic acid supplementation: a systematic review and meta-analysis. Crit Rev Food Sci Nutr. 2021 Oct 21:1-17. doi: 10.1080/10408398.2021.1993781.

  1. Gestational Choline and Fetal Brain Development.

Page 3, line 153. There is a typographical error in the size of the font used: “lysophosphatidylcholine [53]. Choline is a component of sphingomyelin”.

  1. The Current State of Knowledge on the Association Between Maternal Folate and Offspring Cognitive Outcomes.

The meta-analysis by Chen et al. should be incorporated.

  1. Gaps in the Current State of Knowledge and Directions for Future Research.

Page 9, line 433. Although the acronym ASD (Autism Spectrum Disorder) is common, it should be made explicit.

  1. Conclusion.

Page 9, line 469. There is a typographical error and the word “Introduction” should be removed.

References.

Reference 33 is incomplete. References should be checked throughout the manuscript.

Author Response

The manuscript is an interesting narrative review, but requires some considerations.

Title:  You should make it explicit that this is a narrative review.

We have changed the title to make it explicit that the review is a narrative review; see lines 2-3

Prenatal Folate and Choline Levels and Brain and Cognitive Development in Children: A Critical Narrative Review

  1. Introduction.

Page 2, line 84. The search criteria used should be exposed: dates, keywords...

We have included this information on the search criteria on lines 107-115.

  1. Gestational Folate and Fetal Brain Development.

This meta-analysis should have been discussed:

Chen H, Qin L, Gao R, Jin X, Cheng K, Zhang S, Hu X, Xu W, Wang H. Neurodevelopmental effects of maternal folic acid supplementation: a systematic review and meta-analysis. Crit Rev Food Sci Nutr. 2021 Oct 21:1-17. doi: 10.1080/10408398.2021.1993781.

We thank the reviewer for bringing this meta-analysis to our attention. As it was not published in the time frame during which we conducted our literature searches, we had not encountered it. As the Chen et al. meta-analysis did not specifically examine gestational folate and fetal brain development we have not referred to it in this section of the manuscript; however, we have included a discussion of the findings of this meta-analysis in the section on “Maternal Folate and Offspring Cognitive Outcomes”; see lined 309-313.

Gestational Choline and Fetal Brain Development.

Page 3, line 153. There is a typographical error in the size of the font used: “lysophosphatidylcholine [53]. Choline is a component of sphingomyelin”.

This typographical error has been corrected.

  1. The Current State of Knowledge on the Association Between Maternal Folate and Offspring Cognitive Outcomes.

The meta-analysis by Chen et al. should be incorporated.

We thank the reviewer for bringing this meta-analysis to our attention. As it was not published in the time frame during which we conducted our literature search we had not encountered it. Please see out discussion of this paper on lines 309-313..

  1. Gaps in the Current State of Knowledge and Directions for Future Research.

Page 9, line 433. Although the acronym ASD (Autism Spectrum Disorder) is common, it should be made explicit.

We have made this change.

  1. Conclusion.

Page 9, line 469. There is a typographical error and the word “Introduction” should be removed.

We have made these changes.

References.

Reference 33 is incomplete. References should be checked throughout the manuscript.

Reference 33 has been updated. We have checked the references for their accuracy and added the Chen et al reference.

Reviewer 3 Report

This narrative review entitled Prenatal Folate and Choline Levels and Brain and Cognitive Development in Children is really interesting.
In my opinion the text is well written.
I suggest the authors to insert some tables to facilitate the reading of the results.
I also suggest inserting an image that explains the mechanisms of action.
These actions could make the reading of scientific work more engaging.

Author Response

This narrative review entitled Prenatal Folate and Choline Levels and Brain and Cognitive Development in Children is really interesting. In my opinion the text is well written.

We thank the reviewer for their positive comments.

I suggest the authors to insert some tables to facilitate the reading of the results.

We have added Tables 1 and 2, which summarize the animal and human studies that have investigated the associations between folate or choline and neurodevelopmental outcomes.

I also suggest inserting an image that explains the mechanisms of action. These actions could make the reading of scientific work more engaging.

We have included Figures 1 and 2, which provide schemas for the mechanisms of actions for folate and choline.